# Impact of Robotic-Assisted Gait Training in Subacute Spinal Cord Injury Patients on Outcome Measure

**DOI:** 10.3390/diagnostics13111966

**Published:** 2023-06-05

**Authors:** Beata Tarnacka, Bogumił Korczyński, Justyna Frasuńska

**Affiliations:** 1Department of Rehabilitation, Medical University of Warsaw, 02-637 Warsaw, Poland; frasunska@gmail.com; 2Research Institute for Innovative Methods of Rehabilitation of Patients with Spinal Cord Injury, Health Resort Kamień Pomorski, 72-400 Kamień Pomorski, Poland; b.korczynski@u-kp.pl

**Keywords:** robotic therapy, SCI, rehabilitation, Walking Index, disability scales

## Abstract

The improvement of walking ability is a primary goal for spinal cord injury (SCI) patients. Robotic-assisted gait training (RAGT) is an innovative method for its improvement. This study evaluates the influence of RAGT vs. dynamic parapodium training (DPT) in improving gait motor functions in SCI patients. In this single-centre, single-blinded study, we enrolled 105 (39 and 64 with complete and incomplete SCI, respectively) patients. The investigated subjects received gait training with RAGT (experimental S1-group) and DPT (control S0-group), with six training sessions per week over seven weeks. The American Spinal Cord Injury Association Impairment Scale Motor Score (MS), Spinal Cord Independence Measure, version-III (SCIM-III), Walking Index for Spinal Cord Injury, version-II (WISCI-II), and Barthel Index (BI) were assessed in each patient before and after sessions. Patients with incomplete SCI assigned to the S1 rehabilitation group achieved more significant improvement in MS [2.58 (SE 1.21, *p <* 0.05)] and WISCI-II [3.07 (SE 1.02, *p <* 0.01])] scores in comparison with patients assigned to the S0 group. Despite the described improvement in the MS motor score, no progression between grades of AIS (A to B to C to D) was observed. A nonsignificant improvement between the groups for SCIM-III and BI was found. RAGT significantly improved gait functional parameters in SCI patients in comparison with conventional gait training with DPT. RAGT is a valid treatment option in SCI patients in the subacute phase. DPT should not be recommended for patients with incomplete SCI (AIS-C); in those patients, RAGT rehabilitation programs should be taken into consideration.

## 1. Introduction

Spinal cord injury (SCI) is a serious condition limiting a patient’s mobility and physical activity. Additionally, it affects the patient’s functioning and participation in private and social life [1]. Improvement in gait function is a fundamental health priority for individuals with SCI [2,3,4]. A comprehensive rehabilitation program provides the basis for achieving this goal by contributing to the improvement of clinical conditions and increasing the activity and social participation of patients with SCI [5,6].

Recently, the use of robotic devices as a branch of physiotherapy for gait rehabilitation in patients with SCI, as well as for multiple sclerosis, cerebral palsy, and Parkinson’s disease, has been developed [7,8,9]. In the literature, several reports have confirmed its effectiveness, mostly being performed in the chronic phase of disease or as a continuation of rehabilitation programs [4,10,11,12]. A recent review and meta-analysis suggest that robotic-assisted gait training (RAGT) is not only safe and tolerated, but its results showed a positive effect even superior to conventional gait training [13]. Regular locomotor training can also prevent secondary problems associated with physical inactivity in SCI patients. RAGT has been shown to have beneficial effects in reducing pain, spasticity, or improving cardiopulmonary, urinary, and bowel functions [4,11,14,15,16]. The greatest effects of RAGT are observed in individuals with incomplete SCI and in the early stages of this therapy [12]. It is also known that the application of robotic therapies in animal models through functional and repetitive motion and its effects on locomotor support also stimulate neuroplasticity [17]. There are also reports that RAGT has limitations, especially in physiological gait movement performance [18,19].

Studies on the effect of RAGT in patients with SCI have mostly been conducted as single interventions with a small number of participants or as case reports [4,14]. In addition, most studies were performed in chronic SCI stages without analysis of the period since injury [4,10,20]. In the current literature, there are no comparisons between RAGT and dynamic parapodium training (DPT), the methods used in conventional gait therapy in SCI patients. We think that there is a very important aspect to explore, because in many countries, including Poland, robotic rehabilitation is not reimbursed by the state, and for most people, it is an expensive therapy. On the contrary, parapodium is cheaper and reimbursed by the state. 

The main aim of this randomized study was to evaluate the effectiveness of gait training using a robot-driven gait orthosis in patients with SCI in comparison with ground conventional gait training with dynamic parapodium. In most studies concerning the physical improvement and gait quality of RAGT, gait functional scales were used, but other functional measures were used very scarcely. Taking that into consideration, the second purpose of this study was to evaluate the effect of robotic therapy on functional and motor improvement measured by validated clinical and functional scales in patients with subacute SCI. 

## 2. Methods

### 2.1. Data Source and Study Population

This was a single-centre, single-blinded, single-armed study. Patients were recruited through self-selection from all over our country. All participants agreed to participate and signed an informed consent form before the study. The study was approved by the Ethical Board of the District Medical Chamber in Szczecin (Poland) (Nr OIL-SZ/MF/KB/452/05/07/2018; Nr OIL-SZ/MF/KB/450/UKP/10/2018).

The inclusion criteria for the study were as follows: time since injury: from 3 months to 2 years; general condition of the patient: conscious and able to cooperate with a physiotherapist and adapted to an upright position; complete or incomplete SCI (cervical, thoracic, or lumbar) with preserved flexion and extension function at the elbow and wrist; postoperative stabilization with completed bone fusion; absence of contraindications in rehabilitation, such as thrombophlebitis, pulmonary embolism, orthostatic hypotension, epilepsy, and infection; and body weight less than 120 kg and height between 155 cm and 190 cm.

The exclusion criteria for the study were as follows: patients with high complete tetraplegia and very low lumbar spine injury; lack of completed bone fusion after spine surgery; respiratory insufficiency, circulatory insufficiency III and IV New York Heart Association (NYHA) grade; osteoporosis; lower limb shortening of more than 2 cm; the presence of decubitus ulcers; pressure ulcers or skin lesions that may be exacerbated by robotic systems; intensive spasticity (4 points on the Ashworth scale), and the presence of contractures, making it impossible to conduct robotic rehabilitation; pre-existing conditions causing neurological disorders, such as trauma, spinal stroke, multiple sclerosis, and infantile cerebral palsy; symptoms of recurrent autonomic dysreflexia. One patient aged 12 years who met all the inclusion criteria was included in the study after obtaining appropriate consent from his legal guardians.

The therapeutic program consisted of two phases: first, 3 weeks, then, after a 1-week break, 3 weeks in the second phase. The program was conducted six days per week. Simple randomization was used by tossing a coin. A blinded investigator (a physiotherapist that was not involved in the treatment process) was responsible for the group allocation process. Physiotherapists blinded to the aim of the study performed the treatments. Participants were allocated into two groups: (a) the control group (S0) which received conventional gait therapy with DP; and (b) the experimental group (S1), which received RAGT. The S1 group underwent 30 min sessions of RAGT with exoskeleton EKSO-GT (Figure 1, model EKSO 1 by Ekso Bionics, year of manufacture 2014) or Locomat Pro (Figure 2, model LO218 by Hocoma AG, year of manufacture 2014) with the general exercise program and ground gait training; control group S0 also received conventional physiotherapy and 30 min DPT. The dynamic parapodium is a piece of individualized uprighting equipment (a combination of thoracolumbosacral orthosis and hip-knee-ankle-foot orthosis (HKAFO) device of the dynamic type) that allows the patient to stand and walk by swinging the trunk. All sessions were supervised by a trained therapist. All participants from the Locomat group with incomplete SCI started with 60% body weight support and an initial treadmill speed of 1.5 km/h; patients with complete SCI started with 100–90% body weight support. In patients with EKSO-GT, minimum 100 steps were required per session. Patients with a thoracic level of injury were mostly enrolled in the EKSO-GT group; and with a cervical level, in the Locomat group. In cases of muscle pain, patients underwent additional physiotherapy treatment, including classical massage, hydromassage, laser therapy, or dry CO_2_ baths. All patients were evaluated twice by a blinded physiotherapist and physician (not involved in the rehabilitation program at baseline) before the start of therapy and after 7 weeks of therapy. We performed a standard neurological and functional assessment procedure in all patients using scales of proven reliability, validity, and sensitivity to change. In all patients, The American Spinal Cord Injury Association (ASIA) Impairment Scale (AIS) [21], Spinal Cord Independence Measure, version III (SCIM-III) [22], Walking Index for Spinal Cord Injury, version II (WISCI-II) [23], and Barthel Index (BI) [24] assessments were conducted.

The detailed study flowchart is shown in Figure 3.

### 2.2. Statistical Methods

Continuous variables with non-normal distributions are presented as the median and interquartile range (IQR), while continuous variables with normal distribution are presented as mean with one standard deviation. The distribution of the variables was checked using the Shapiro–Wilk test. Nominal variables are presented as the number of observed cases and percentage n (%). Variables were compared between rehabilitation groups using the *t*-test, Mann–Whitney test, χ^2^ test, or Fisher’s exact test for nominal variables. The results were considered significant at *p* < 0.05. Within-group differences were estimated using the *t*-test or Wilcoxon Cox test for paired observations (variables measured at baseline and after rehabilitation). Differences between rehabilitation groups regarding the changes in outcome measures at the end of the study relative to the outcome before rehabilitation were analysed using linear regression (mean magnitude of change between groups, SE). The following factors were included in the analysis as confounders: baseline value of the analysed variable, sex, age, and ASIA as covariables in the model.

Statistical analysis, data preparation, and visualization were performed using the R software (R Core Team (2021)) [25], supplemented with the following packages: readxl [26], ggplot2 [27], qwraps2 [28], rmarkdown [29], ggpubr [30], Huxtable [31], tidyverse [32].

## 3. Results

### 3.1. Characteristics of the Investigated Group

Finally, we enrolled 105 patients with SCI: 39 with complete and 64 with incomplete injury; 72 in the RAGT and 33 in the control group (Figure 3). Patients were admitted to Health Resort Kamień Pomorski S.A. Research Institute for Innovative Methods of Rehabilitation of Patients with Spinal Cord Injury in Kamień Pomorski from 2018 to 2021. The epidemiological situation caused by the COVID-19 pandemic limited 16 patients from completing the rehabilitation program from the S0 group. The disproportion of the number enrolled to the S0 group was also related to the fact that patients who drew the group without RAGT dropped out of rehabilitation at their own request. All patients participating in the study had neurological impairment following SCI: 39 (39%) patients were scored as AIS-A and 64 (61%) as AIS-B, C, or D. The characteristics of the S1 and S0 groups are presented in Table 1.

### 3.2. Functional and Neurological Improvement Depending on the Type of Rehabilitation

We aimed to compare the effects of rehabilitation with RAGT (S1) vs. DPT (S0) on functional and neurological patients’ status. We assessed how the WISCI-II, SCIM-III, motor index from AIS Motor Scores (MS), and BI scale changed after treatment in relation to baseline, and analysed these changes between intervention groups. Figure 4 presents WISCI-II, SCIM-III, MS, and BI scale changes in investigated interventions in all patients, in complete and incomplete subgroups. A statistically significant improvement in the WISCI-II (*p <* 0.001) and MS subscales (*p <* 0.05) was observed in all and incomplete patients from the S1 group compared with S0 (*p <* 0.01, respectively). Subsequently, we performed linear regression analysis with selected covariates to confirm if the observed effect was due to rehabilitation type. We assessed the rehabilitation type effect adjusted to the baseline level of parameter and SCI degree (incomplete SCI vs. complete SCI) (Table 2).

Regression analysis showed that RAGT was associated with significantly greater change in the WISCI-II scale parameter, regardless of the covariables included in the model. The adjusted difference between the S1 and S0 rehabilitation groups averaged 1.94 (SE = 0.63, *p* < 0.01); the analysis did not show the interaction between the type of rehabilitation and SCI scales (Table 2).

Regression analysis for MS showed a significant association between the change in MS value after rehabilitation and rehabilitation type (S0 vs. S1). Patients assigned to S1 rehabilitation had on average a greater incremental value of 1.73 (SE = 0.74, *p <* 0.5) compared to group S0. In addition, the interaction model showed that the relationship between rehabilitation type (S0 vs. S1) and MS change was modulated by SCI severity (Table 2).

Regression analysis confirmed that the SCIM-III and BI score changes were not related to the type of rehabilitation assigned to the patient, either in the simple analysis or adjusted for the other covariables in the model (Table 2).

The changes in the analysed parameters (MS, WISCI-II, SCIM-III, and BI) concerning age and time since the accident were measured. The analysis revealed no significant association between these variables (Table 3).

### 3.3. Effect of Rehabilitation on the Functional and Neurological Scales in Patients with Incomplete Spinal Cord Injury (AIS-B, C, and D)

In Table 4, a comparison between patients with incomplete and complete SCI is provided. Patients from the S1 group with incomplete SCI had a greater benefit from rehabilitation than patients with complete SCI (MS, *p <* 0.001). The same pattern was observed in the group of patients who underwent S0 rehabilitation; however, the differences did not reach statistical significance (MS, *p* = 0.08). No significant differences were observed between the other parameters related to the performance in complete and incomplete SCI.

To assess if patients with incomplete SCI benefited from rehabilitation, we compared the final and initial values of WISCI-II, SCIM-III, BI, and MS. All analysed parameters related to the functional and neurological performance of patients (WISCI-II, SCIM-III, BI, and MS) showed a significant improvement, regardless of rehabilitation type. Results for S0 and S1 are shown separately in Figure 5.

Then, we aimed to assess if RAGT gave a superior effect compared to DPT in patients with incomplete SCI. Regression analysis showed that the change in the WISCI-II parameters after rehabilitation relative to baseline was significantly related to the type of rehabilitation (S0 vs. S1) in incomplete SCI patients. The relationship was shown by both the simple regression model and the model adjusted for the covariates. The adjusted difference between the groups was on average [3.07 (SE 1.02, *p <* 0.01].

The regression analysis presented in Table 5 also shows significant differences between the S1 and S0 rehabilitation types, regardless of the covariates in a model. In each model, patients assigned to the S1 rehabilitation group had a larger MS change [2.58 (SE 1.21, *p* < 0.05)] than patients assigned to the S0 group.

## 4. Discussion

The present study compared the effect of RAGT vs. DPT in SCI patients. The results of our study show that rehabilitation treatment using RAGT and DPT may improve outcomes in those patients, but patients in the RAGT group achieved significantly better WISCI-II and MS final scores. 

This was a study with simple randomization in a large SCI group of patients with a defined injury period from the acute to subacute phase following SCI. Most previous reports have involved case studies or retrospective analyses. Furthermore, DPT has not been previously compared with RAGT therapy. This is a very important clinical issue, because in Poland, as in many countries, the rehabilitation of patients with SCI stops a few months after injury, with dynamic parapodium being prescribed by doctors or physiotherapists as the only device for gait therapy. Many patients in Poland buy RAGT therapy sessions on their own, hoping for very fast neurological improvement, even if they suffer from a complete SCI. We did not analyse cost–benefit aspects, but we think, that the aspect of RAGT being economically rewarding is relevant in terms of health policies. 

A comparative analysis of the investigated group and control group showed a significantly greater effect of RAGT on MS and WISCI-II parameters compared to S0. This study confirmed the improvement in these parameters previously reported in the literature, both in patients using exoskeletons alone and in patients using Lokomat alone [12,33,34,35]. Alshram et al., in a meta-analysis concerning the impact of RAGT devices such as Lokomat, found that this therapy did not always improve MS scores [16]. We observed clinical improvement in both the control and experimental study groups; however, there were greater effects (in functional gait parameters) in the RAGT group. These results are consistent with reports from the literature [14,15,16,34,36]. In a study by Labruyère et al., no such relationship was observed [37]; however, the study included a small group of only nine patients with incomplete SCI (AIS-C and D). In the meta-analysis performed by Fang et al., the MS score significantly increased in favour of RAGT in both randomized controlled trials (RCTs) and non-RCTs [14]. 

Despite the described improvement in the MS motor score in the studied groups, no progression between grades of AIS (A to B to C to D) was observed. Perhaps this was related to a lack of changes in upper limb muscle strength or/and sensory disturbances. 

The results of the regression analysis confirmed previous observations regarding the relationship between rehabilitation type and SCIM-III score; however, the observed improvement in SCIM-III parameters in both groups of patients in our study did not reach statistical significance. According to the literature, the highest rate of improvement in the SCIM-III index is observed in the first 3 months after SCI, and the rate of clinical improvement in this scale decreases thereafter [37]. Thus, the highest improvement in this parameter occurs when patients are in the early neurological rehabilitation stages. All patients included in our study were after such conventional gait rehabilitation, and were within an average of 15 months following SCI. In our study, no improvement in SCIM-III may be related to its slower time-related functional improvement progression in SCI patients. Furthermore, SCIM-III is not the primary test for gait measurement in SCI [38]. In our study, SCIM-III was not a useful tool for functional outcome measures. SCIM-III results were similar to those of other studies [34,38]. However, it was difficult to compare them unambiguously owing to the methodological differences described above. In the literature, this kind of outcome measure was also not frequently used.

The lack of a statistically significant improvement in BI assessment in our study may indicate that this parameter has little clinical utility in the evaluation of patients with SCI [39]. BI analyses 10 basic activities of daily living (ADLs), and does not simultaneously assess important functions for patients with SCI, especially for gait or wheelchair transfer [24].

Most of the clinical studies concerning RAGT in SCI patients used different protocols in terms of treatment, time of sessions, frequency, and criteria for progression, as well as different study methods and designs, making data poorly comparable. We investigated patients in the acute and subacute phases of SCI. It may be difficult to compare our results with other available similar studies because of some methodological differences [33,34,35]. There is a lot of investigation performed on small samples and in a chronic stage of the disease. For instance, the time since injury was up to 33 years after SCI in some studies [33], whereas it was no longer than 36 months in our study. According to the literature, the period of the injury affects the rate of improvement. However, this was not confirmed by the results of our study. In a study by Benito-Penalva et al., the highest rate of improvement was observed in the first 6 months after SCI [40]. Benito-Penalva’s study compared SCI rehabilitation with other robotic forms (Lokomat and Gait Trainer GT I systems) without a control group [40]. A different result was reported by Zieriacks et al. in their study [30]. However, this study mainly included patients with incomplete injury without AIS-B (97 patients), and patients with AIS-A (24 patients) who had lesions from the conus medullaris to the cauda equina with zones of partial preservation and existing motor function of the hip and knee extensor and flexor muscle groups to operate the exoskeleton. In addition, the patients recruited for that study had an injury in a period between 1 year and 33 years after SCI, which excludes the possibility of comparing our study with the results [33].

Our treatment protocol consisted of 6 weeks of therapy sessions administered 6 times per week. In some patients, especially with complete SCI, it was a long period, and in some patients, we observed a high level of fatigue. Some patients would have needed longer recovery, i.e., breaks between sessions. We did not test these patients for endurance, because that was not our goal, but it seems that two to three sessions per week would be completely absolutely sufficient for complete SCI patients. 

A greater benefit from robotic rehabilitation was observed in patients with incomplete SCI than in patients with complete SCI. This finding is consistent with previous speculations. A major problem in the assessment of the efficacy of rehabilitation is related to the patient’s level of disability and injury. It is also known that RAGT effectiveness may be related to the degree of disability. We performed our study including patients with complete SCI. We did not expect a neurological improvement in these patients, but in functional scales such as SCIM-III or BI, which we did not observe to our surprise. In the literature, the SCIM-II score improvement even during the 1-year follow-up was observed in patients with complete paraplegia [38]. Detailed analysis performed in patients with incomplete injury (AIS-B, C, D) showed significant improvement in all functional parameters used in the study (WISCI-II, MS, BI, and SCIM-III) in both the S0 and S1 groups. However, regression analysis showed that RAGT was associated with superior improvements in the MS and WISCI-II scales, which is the most relevant result of this study. The WISCI-II scale is the main parameter used to assess gait in patients with SCI and is a good marker for assessing locomotor ability in patients with SCI [41]. A significant number of results reported in the literature also investigated patients with incomplete SCI; however, there are some differences in methodology to compare, as described above [31,32,33]. There were gaps in the reports analysing RAGT in patients with incomplete AIS grades. In particular, we wanted to draw attention to the AIS-B group, which is often overlooked in previous studies, although it belongs to the incomplete SCI, and a higher degree of conversion to better neurological levels than in the complete SCI (AIS-A) was observed in the literature [41,42]. Thus, the present study illustrates a certain advantage of combined therapy using robotic training in addition to conventional therapy over using different single forms of physiotherapy. The merit of RAGT is that as devices, they provide complete strain relief and repetitive and systematic locomotion, as well as stimulating neuroplasticity [4,17]. However, in addition to the above-described effect of RAGT on locomotor training of patients with SCI, we emphasize that conventional therapy, with no body weight support, may influence clinical improvement, as observed in our patients in the S0 group. Indeed, the rate of this improvement does not increase as significantly as in the case of RAGT therapy. However, the influence of gait re-education based only on physiotherapy in patients with incomplete SCI can be seen. We think that the best candidate for RAGT is a patient with an AIS-C grade, especially when improving gait function is the primary functional goal in these patients and a priority for recovery and quality of life improvement following SCI [4].

The results reported by reviews and meta-analyses show no significant differences between devices used in RAGT therapy for SCI patients [42,43]. We used two RAGT devices: Locomat is a stationary robotic device based on the treadmill and exoskeleton EKSO-GT, and therefore, we did not want to exclude patients with poor upper limb strength. The comparison between those two devices was also not related to our study subjects. This can raise the question of comparability of the results obtained from rehabilitation training conducted with different robotic devices. In the literature, opinions can be found that end-effector practice can provide higher movement variability during gait training and stimulate the neuroplasticity mechanism of recovery at the central nervous system level more significantly compared to exoskeleton training [44,45]. Most of our patients from the incomplete and complete thoracic SCI groups were assigned to the EKSO group, and most patients with incomplete cervical SCI were assigned to the Locomat group because of upper extremity paresis. We decided not to compare those devices because of the small samples for further analysis. Moreover, there is currently no definitive answer in the literature as to what kind of RAGT device is more effective in SCI patients, and this was also not the subject of our research. 

In previous studies, the reported RAGT limitations mostly concerned the occurrence physiological gait movements, such as the abnormal sensory stimuli created by a strap used to fix the patient’s lower limbs to the robot; the decrease in muscle activities that produce stability and propulsive force; passively induced movements; the occurrence of sagittal plane lower limb movements; the lack of movements of the trunk and pelvis; or the absence of an effective weight shift [18]. The other study conducted by Bae et al. aimed to analyse the effects of RAGT on foot pressure to determine an effective training protocol for patients with incomplete SCI [19]. The authors found that during robotic therapy, lower peak foot pressure and shorter stance phase duration are seen, and they concluded that this fact can limit gait pattern improvement in SCI patients undergoing robotic therapy [19]. However, we must emphasize that these were preliminary studies performed on a very small number of patients (four) with low thoracic and high lumbar lesions. In a parapodium device patient may have active muscles, such as the erector spine, gluteus medius, biceps femoris, etc; the patient is also able to activate proprioception. However, parapodium has also limitations, such as the great effort required by the patient and the therapy being very monotonous for the patient, which can greatly reduce the motivation to exercise. It is for this reason that this rehabilitation equipment is often not used, despite being purchased by the patient.

This study included patients from different parts of our country; however, it was conducted in a single centre. A deeper analysis of this problem may require multicentre or international studies.

All patients wanted to be allocated to the RAGT group; hence, they often withdrew from continuing the study when assigned to a group without RAGT, leading to the smaller sample size of the control group. The inclusion of these parameters may influence the results of this study.

## 5. Conclusions

RAGT is a useful gait therapy, especially when combined with conventional rehabilitation in patients with SCI. It significantly improves functional and locomotor parameters (in patients with incomplete SCI). DPT should not be recommended for patients with incomplete SCI, especially in patients with AIS-C; those patients should be taken to RAGT rehabilitation programs. The observed progress in robotic-assisted neurorehabilitation is a promising gait therapy method. Therefore, there is a need for further research and the development of more individualized and integrated assistive technologies for functional locomotion in patients with SCI.

## Figures and Tables

**Figure 1 diagnostics-13-01966-f001:**
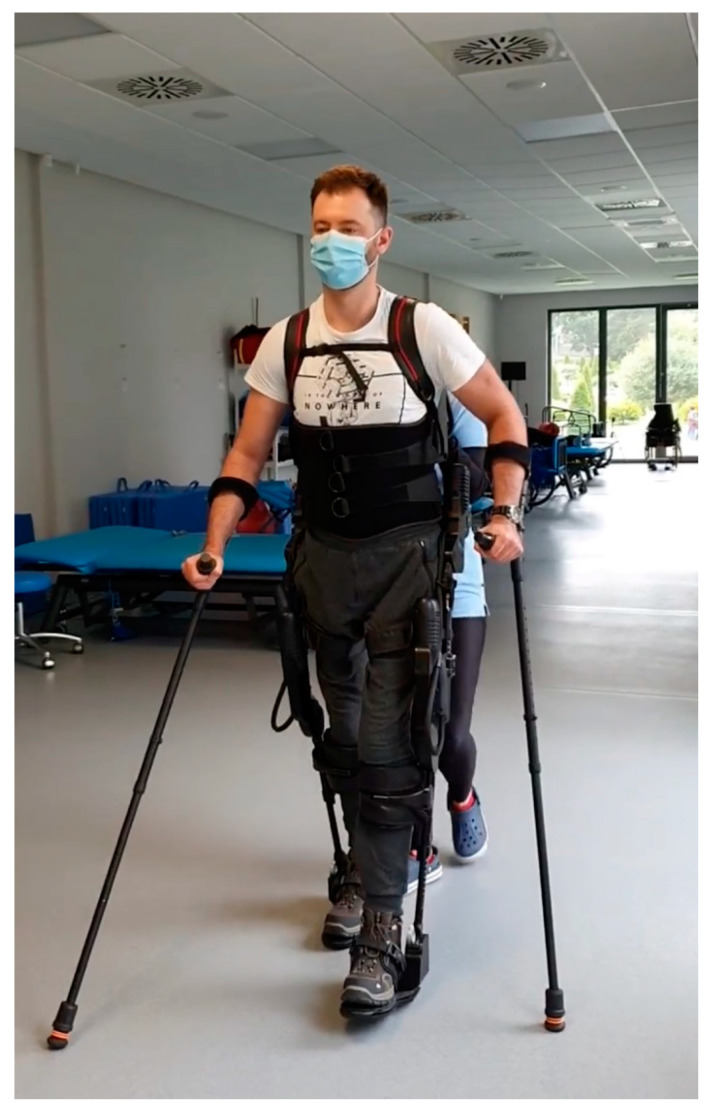
Patient with spinal cord injury undergoing exoskeleton EKSO-GT, model EKSO 1.

**Figure 2 diagnostics-13-01966-f002:**
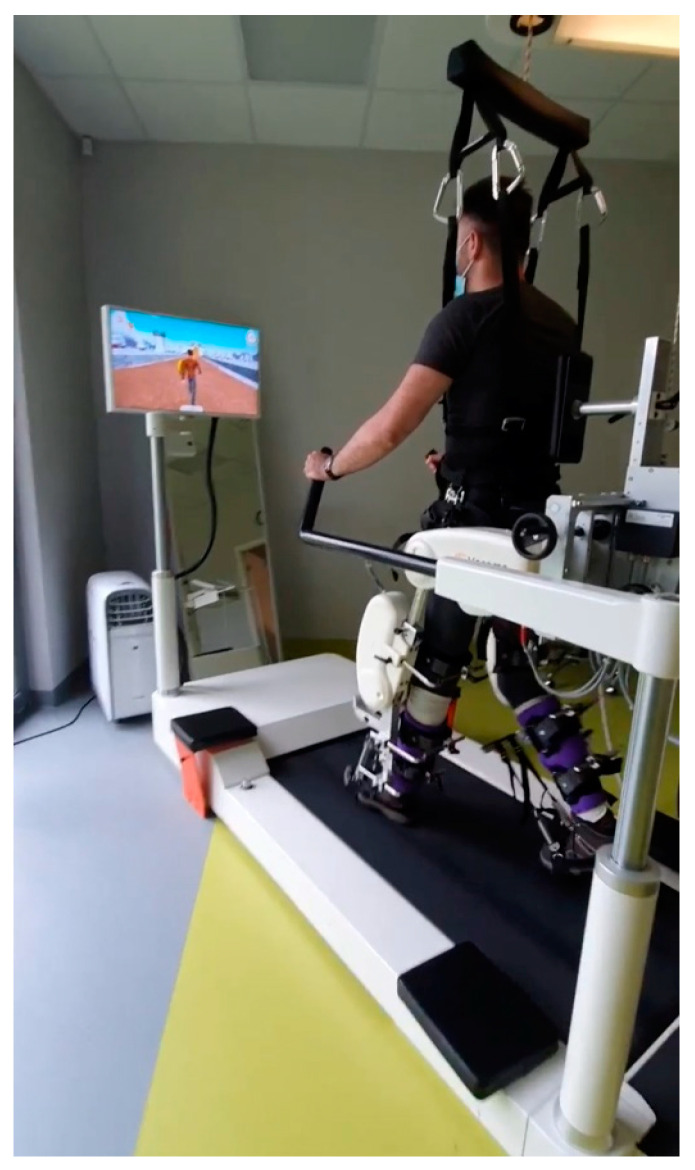
Patient with spinal cord injury undergoing Lokomat-Pro, model LO218.

**Figure 3 diagnostics-13-01966-f003:**
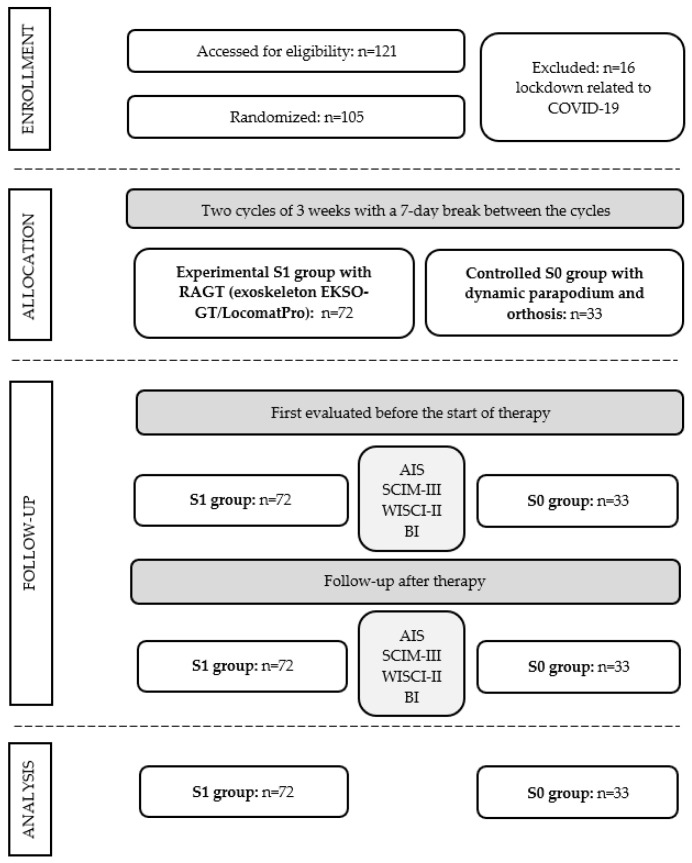
Flowchart of patients’ recruitment.

**Figure 4 diagnostics-13-01966-f004:**
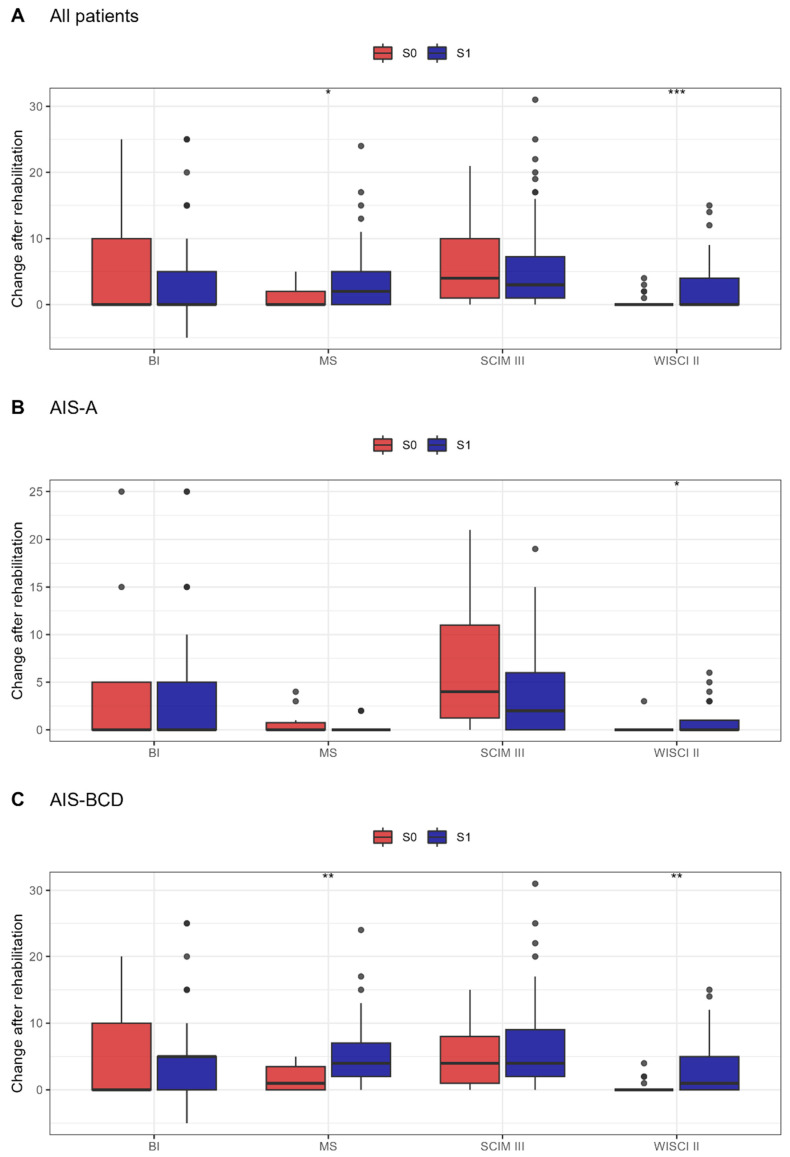
(**A**–**C**) Parameter changes after rehabilitation relative to the initial value between investigated groups: rehabilitation with DPT (S0) and rehabilitation with RAGT (S1) in all (**A**), in complete (**B**), and incomplete (**C**) subgroups; * *p* ≤ 0.05, ** *p* ≤ 0.01, *** *p* ≤ 0.001.

**Figure 5 diagnostics-13-01966-f005:**
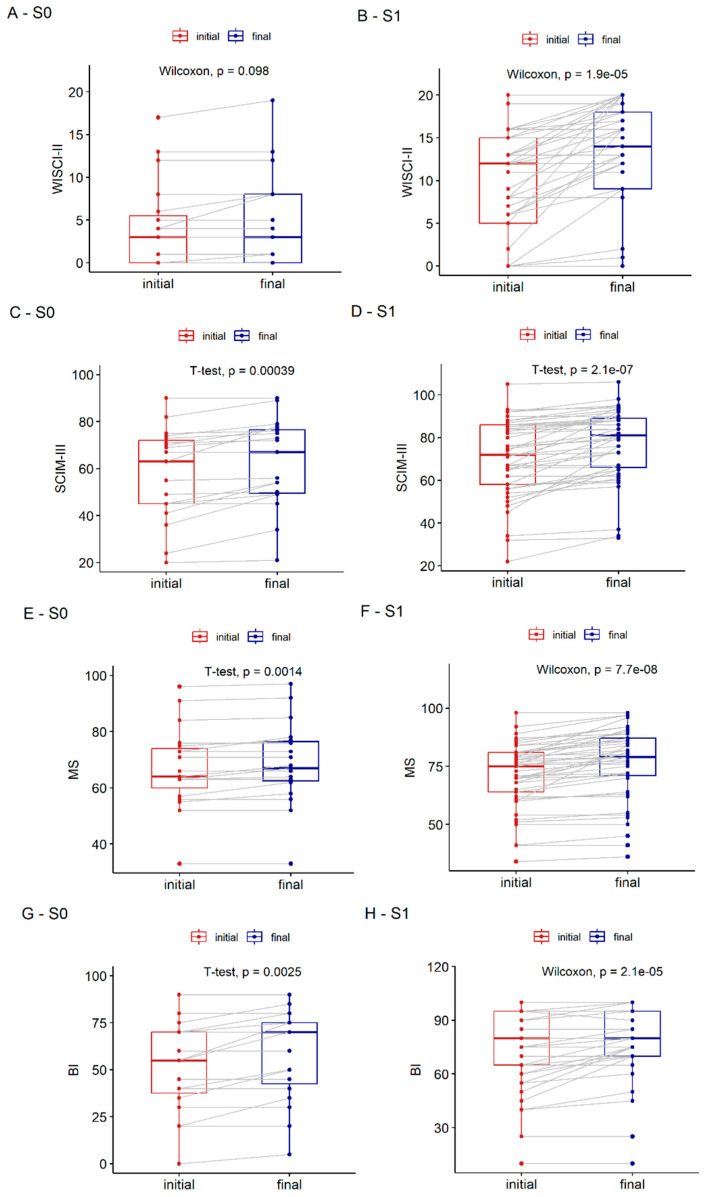
(**A**–**F**) Comparison of final and initial parameter values (WISCI-II, SCIM-III, MS and BI) in S0 and S1 group patients with incomplete SCI: (**A**) WISCI-II rehabilitation type S0 (*p* = 0.1); (**B**) WISCI-II rehabilitation type S1 (*p <* 0.001); (**C**) SCIM-III rehabilitation type S0 (*p <* 0.001); (**D**) SCIM-III rehabilitation type S1 (*p <* 0.001); (**E**) MS rehabilitation type S0 (*p* = 0.001); (**F**) MS rehabilitation type S1 (*p <* 0.001); (**G**) BI rehabilitation type S0 (*p* = 0.003); (**H**) BI rehabilitation type S1 (*p <* 0.001).

**Table 1 diagnostics-13-01966-t001:** Characteristics of the investigated group.

Group	S0 (N = 33)	S1 (N = 72)	*p*-Value
Sex			
Women	5 (15.15%)	14 (19.44%)	0.797 ^1^
Men	28 (84.85%)	58 (80.56%)	
Cause of injury			
Vehicle accident	13 (39.39%)	24 (33.33%)	0.067 ^2^
Fall < 1 m	2 (6.06%)	4 (5.56%)	
Fall > 1 m	9 (27.27%)	24 (33.33%)	
Dive	1 (3.03%)	4 (5.56%)	
Body crushing	4 (12.12%)	0 (0.00%)	
Others	4 (12.12%)	16 (22.22%)	
AIS			
A	14 (42.42%)	27 (37.50%)	0.068 ^2^
B	4 (12.12%)	7 (9.72%)	
C	11 (33.33%)	13 (18.06%)	
D	4 (12.12%)	25 (34.72%)	
Level of neurological impairment			
Cervical	7 (21.21%)	17 (23.61%)	0.794 ^1^
Thoracic	17 (51.52%)	32 (44.44%)	
Lumbar	9 (27.27%)	23 (31.94%)	
Age			
Median (IQR)	36.5 (20.5)	35.5 (16.75)	0.767 ^3^
Time from accident (months)			
Median (IQR)	13 (13)	13 (10)	0.594 ^3^

^1^ Chisquared Test; ^2^ Fisher exact Test; ^3^ Mann–Whitney Test; Abbreviations: AIS: American Spinal Cord Injury Impairment Scale (grades A, B, C and D); IQR: interquartile range; N—number of respondents; *p*—statistical level; S0: control group; S1: experimental group.

**Table 2 diagnostics-13-01966-t002:** Changes of Walking Index for Spinal Cord Injury—version II, Spinal Cord Independence Measure—version III, Scale Motor Scores, and Barthel Index in regression models.

	S0 (N = 33)			S1 (N = 72)			Unadjusted Model	Adjusted Model
	Baseline, Mean ± SD	Final Mean ± SD	Change from Baseline, Mean ± SD	Baseline, Mean ± SD	Final, Mean ± SD	Change from Baseline, Mean ± SD	S0–S1 Difference (SE)	S0–S1 Difference ^$^ (SE)
WISCI-II	2.73 ± 4.69	3.09 ± 4.97	0.36 ± 0.96	6.54 ± 6.52	8.83 ± 7.61	2.29 ± 3.43	1.93 (0.61) **	1.940 (0.63) **
SCIM-III	57.97 ± 17.10	63.61 ± 15.90	5.64 ± 5.43	66.97 ± 15.88	72.65 ± 15.39	5.68 ± 6.67	0.044 (1.33)	1.113 (1.23)
MS	60.09 ± 14.74	61.30 ± 14.97	1.21 ± 1.71	63.96 ± 14.91	67.16 ± 17.30	3.20 ± 4.54	1.99 (0.82) *	1.73 (1.12) *
BI	56.21 ± 21.25	61.21 ± 19.53	5.00 ± 6.96	70.35 ± 19.94	75.28 ± 17.97	5.00 ± 7.12	0 (1.49)	−0.26 (1.5)

Abbreviations: MS: ASIA (American Spinal Cord Injury) Scale Motor Scores; N—number of respondents; SCIM-III: Spinal Cord Independence Measure, version III; S0: control group; S1: study group; SD: standard deviation; WISCI-II: Walking Index for Spinal Cord Injury, version II; S0: control group; S1: experimental group; SD: standard deviation; ** *p* < 0.01; * *p* < 0.05; (SE—standard error); ^$^ regression analysis, adjusted for covariates: baseline value of parameter, SCI (incomplete vs. complete).

**Table 3 diagnostics-13-01966-t003:** The effect of age and time since the accident on the different variables studied (MS, WISCI-II, SCIM-III, and BI).

Parameter	MS_Change	pval	WISCI_Change	pval	SCIM_Change	pval	BI_Change	pval
Age	−0.04	0.56	−0.11	0.15	0.14	0.05	0.03	0.66
Time since the accident	0.08	0.29	0.06	0.41	−0.007	0.92	0.01	0.89

Abbreviations: BI: Barthel Index; MS: ASIA (American Spinal Cord Injury) Scale Motor Scores; pval: (*p*-value) statistical level; SCIM-III: Spinal Cord Independence Measure, version III; WISCI-II: Walking Index for Spinal Cord Injury, version II.

**Table 4 diagnostics-13-01966-t004:** Change in outcome parameters related to dexterity and independence between patients with incomplete (AIS-B, C, and D) and complete (AIS-A) SCI in RAGT and control groups.

		**AIS-A (*n* = 14)**	**AIS-B, C, and D (*n* = 19)**	
**S0 N = 33**		**Median (IQR)**	**Median (IQR)**	***p*-Value**
	WISCI-II	0 (0–0)	0 (0–0)	0.32
	SCIM-III	4 (11–1.25)	4 (8–1)	0.45
	MS	0 (0.75–0)	1 (3.5–0)	0.08
	BI	0 (5–0)	0 (10–0)	0.60
		**AIS-A (*n* = 27)**	**AIS-B, C, and D (*n* = 45)**	
**S1 N = 72**		**Median (IQR)**	**Median (IQR)**	***p*-Value**
	WISCI-II	0 (1–0)	1 (5–0)	0.63
	SCIM-III	2 (6–0)	4 (9–2)	0.06
	MS	0 (0–0)	4 (7–2)	<0.001
	BI	0 (5–0)	5 (5–0)	0.64

Abbreviations: AIS: American Spinal Cord Injury Impairment Scale (grades A, B, C, and D); IQR: interquartile range; MS: ASIA (American Spinal Cord Injury) Scale Motor Scores; *n*: number of respondents; *p*: statistical level; RAGT: robotic-assisted gait training; SCI: spinal cord injury; SCIM-III: Spinal Cord Independence Measure, version III; S0: control group; S1: experimental group; SD: standard deviation; WISCI-II: Walking Index for Spinal Cord Injury, version II.

**Table 5 diagnostics-13-01966-t005:** Type of rehabilitation on functional and neurological parameter changes. Analysis of patients with incomplete spinal cord injury (AIS-B, C, and D).

	S0 (N = 19)			S1 (N = 45)			Unadjusted Model	Adjusted Model
	Baseline, Mean ±SD	Final Mean ± SD	Change from Baseline, Mean ± SD	Baseline, Mean ± SD	Final, Mean ± SD	Change from Baseline, Mean ± SD	S0–S1 Difference (SE)	S0–S1 Difference ^$^ (SE)
WISCI-II	4.05 ± 5.09	4.53 ± 5.47	0.47± 1.07	9.42 ± 6.42	12.47 ± 6.92	3.04± 3.95	2.57 (0.92) **	3.07 (1.02) **
SCIM-III	57.58 ± 19.55	62.53 ± 18.69	4.95 ± 4.96	70.56 ± 18.05	77.24 ± 16.75	6.69± 7.30	1.74 (1.8)	2.29 (1.799)
MS	67.05 ± 14.45	68.68 ± 14.50	1.63± 1.89	71.47 ± 14.08	76.46 ± 15.46	4.99± 4.93	3.36 (1.17) **	2.583 (1.21) *
BI	52.37 ± 23.94	57.89 ± 23.65	5.53 ± 6.85	75.33 ± 21.41	80.33 ± 19.61	5 ± 6.83	−0.53 (1.87)	1.15 (1.86)

Abbreviations: MS: ASIA (American Spinal Cord Injury) Scale Motor Scores; N: number of respondents; SCIM-III: Spinal Cord Independence Measure, version III; S0: control group; S1: experimental group; SD: standard deviation; WISCI-II: Walking Index for Spinal Cord Injury, version II; ** *p* < 0.01; * *p* < 0.05. (SE—standard error); ^$^ Regression analysis, adjusted for covariates: initial values of parameters, sex, age, level of damage (AIS).

## Data Availability

The datasets generated during the present study are available from the corresponding author upon request.

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
