# Peer review of "Impact of Robotic-Assisted Gait Training in Subacute Spinal Cord Injury Patients on Outcome Measure"

_diagnostics, 2023, doi:10.3390/diagnostics13111966_

Round 1
Reviewer 1 Report
Overall I think this is a generally well written study with a sound methodology and interesting results.
For some reason the figures didn't show up in the submitted manuscript. Those at very least need to be added so that they can be evaluated prior to accepting this submissions.
I don't think all of the data should be presented as the regression results and changes. The original values on each of the scales need to be reported, due to floor and ceiling effects as well as understanding the starting point of the specific patient population.
I'm also disappointed that only 2 evaluations were made for each participant. But I there isn't anything the authors can do about that at this point.
Reviewer 2 Report
The Authors performed the prospective study with the participation of 105 spinal cord injuries (SCI) patients, 39 and 64 with complete and incomplete injuries. The number of samples is impressive making the study highly reliable.
Investigated subjects received gait training with RAGT [experimental S1-group) and without dynamic parapodium training (DPT), control S0-group]. They received six training sessions per week over seven weeks. The comparative study design is proper.
The American Spinal Cord Injury Association Impairment Scale Motor Score (MS), Spinal Cord Independence Measure, version-III (SCIM-III), Walking Index for Spinal Cord Injury, version-II (WISCI-II) and Barthel Index (BI) were performed in each patient before and after therapeutic sessions. The methods of the assessment are of high credibility.
The results proved that RAGT in comparison with conventional gait training and DPT in SCI-patients significantly improved functional parameters in SCI-patients in acute and subacute phase.
The article is interesting but includes some flaws.
1. The title too much resembles the titles of the other original articles on the similar topic as follows:
Shin, J. C., Kim, J. Y., Park, H. K., & Kim, N. Y. (2014). Effect of robotic-assisted gait training in patients with incomplete spinal cord injury. Annals of rehabilitation medicine, 38(6), 719–725. https://doi.org/10.5535/arm.2014.38.6.719 (already cited)
Shahin AA, Shawky SA, Rady HM, Effat DA, Abdelrahman SK, et al. (2017) Effect of Robotic Assisted Gait Training on functional and psychological improvement in patients with Incomplete Spinal Cord Injury. J Nov Physiother Phys Rehabil 4(3): 083-086. DOI: 10.17352/2455-5487.000053 (should be cited and discussed)
I suggest to slightly modify the current title.
2. Keywords need to be corrected to present the content of the article entirely, do not use too many shorts.
3. The flow chart of the study design in Figure 1 would be cordially invited.
4. The tables are not in accordance with the MDPI style. Names of the statistic tests should be included in the other abbreviations in order. P-values are unnecessary too long.
5. The following articles should be upgraded in the List of references and their content discussed:
Fang, C. Y., Tsai, J. L., Li, G. S., Lien, A. S., & Chang, Y. J. (2020). Effects of Robot-Assisted Gait Training in Individuals with Spinal Cord Injury: A Meta-analysis. BioMed research international, 2020, 2102785. https://doi.org/10.1155/2020/2102785
Li R, Ding M, Wang J, et al. Effectiveness of robotic-assisted gait training on cardiopulmonary fitness and exercise capacity for incomplete spinal cord injury: A systematic review and meta-analysis of randomized controlled trials. Clinical Rehabilitation. 2023;37(3):312-329. doi:10.1177/02692155221133474
Anas R. Alashram, Giuseppe Annino, Elvira Padua, Robot-assisted gait training in individuals with spinal cord injury: A systematic review for the clinical effectiveness of Lokomat, Journal of Clinical Neuroscience, Volume 91, 2021, Pages 260-269, https://doi.org/10.1016/j.jocn.2021.07.019
6. The Discussion should start from the short presentation of the most important findings.
7. The editorial content of the manuscript leaves much to be desired, starting from the title page with affiliations and ending with references that are not ordered in accordance with the MDPI requirements.
8. Photographs presenting the methodological principles would increase the manuscript's originality.
Sometimes difficult to understand.
Author Response
JSCM-D-22-00122
“Effect of Robotic Assisted Gait Training on Changes in Clinical Status
of Patients with Spinal Cord Injury”
The table describing actions taken upon requests of the Reviewers:
|
Remarks of the First Reviewer
|
||
|
N.
|
The Reviewer wrote: |
Authors’ actions: |
|
1 |
Overall I think this is a generally well written study with a sound methodology and interesting results.
|
Thank you for your critical comments, which helped us greatly improve the manuscript. The enhanced manuscript has incorporated the reviewers’ comments below. The manuscript was revised to look for appealing and the language was enhanced to be more formal and appropriate.
|
|
2 |
For some reason the figures didn’t show up in the submitted manuscript. Those at very least need to be added so that they can be evaluated prior to accepting this submissions.
|
I don’t know why the sent drawing files with figures did not arrive. I will of course send them again. |
|
3 |
I don’t think all of the data should be presented as the regression results and changes. The original values on each of the scales need to be reported, due to floor and ceiling effects as well as understanding the starting point of the specific patient population.
|
Parameter change after rehabilitation from baseline was a study endpoint, that was used to evaluate effect of intervention. Regression analysis was performed for unbalanced groups to ensure that the effect of rehabilitation significantly affects the magnitude of the parameter change, regardless of possible confounders. As suggested, tables 2, 3, 4 and 7 have been changed to show the original values, not just the amount of the change.
|
|
4 |
I’m also disappointed that only 2 evaluations were made for each participant. But I there isn’t anything the authors can do about that at this point.
|
Thank you for your valuable advice. I will use the comments in my planned future research. |
|
Remarks of the Second Reviewer
|
||
|
N.
|
The Reviewer wrote: |
Authors’ actions: |
|
5 |
The Authors performed the prospective study with the participation of 105 spinal cord injuries (SCI) patients, 39 and 64 with complete and incomplete injuries. The number of samples is impressive making the study highly reliable. Investigated subjects received gait training with RAGT [experimental S1-group) and without dynamic parapodium training (DPT), control S0-group]. They received six training sessions per week over seven weeks. The comparative study design is proper. The American Spinal Cord Injury Association Impairment Scale Motor Score (MS), Spinal Cord Independence Measure, version-III (SCIM-III), Walking Index for Spinal Cord Injury, version-II (WISCI-II) and Barthel Index (BI) were performed in each patient before and after therapeutic sessions. The methods of the assessment are of high credibility. The results proved that RAGT in comparison with conventional gait training and DPT in SCI-patients significantly improved functional parameters in SCI-patients in acute and subacute phase. The article is interesting but includes some flaws.
|
Thank you for your remarks and your critical comments, which helped us greatly improve the manuscript. The enhanced manuscript has incorporated the reviewers’ comments below. |
|
6 |
1. The title too much resembles the titles of the other original articles on the similar topic as follows: Shin, J. C., Kim, J. Y., Park, H. K., & Kim, N. Y. (2014). Effect of robotic-assisted gait training in patients with incomplete spinal cord injury. Annals of rehabilitation medicine, 38(6), 719–725. https://doi.org/10.5535/arm.2014.38.6.719 (already cited) Shahin AA, Shawky SA, Rady HM, Effat DA, Abdelrahman SK, et al. (2017) Effect of Robotic Assisted Gait Training on functional and psychological improvement in patients with Incomplete Spinal Cord Injury. J Nov Physiother Phys Rehabil 4(3): 083-086. DOI: 10.17352/2455-5487.000053 (should be cited and discussed) I suggest to slightly modify the current title.
|
The title of the article has been improved in accordance with the comments. The new title reads: Impact of the Robotic Assisted Gait Training in subacute spinal cord injured patients on outcome measure |
|
7 |
2. Keywords need to be corrected to present the content of the article entirely, do not use too many shorts.
|
As suggested, the keywords has been improved. The new keywords reads: robotic therapy, SCI, rehabilitation, walking index, disability scale.
|
|
8 |
The manuscript is accompanied by a flowchart as Figure 1. |
The manuscript is accompanied by a flowchart as Figure 1. |
|
9 |
4. The tables are not in accordance with the MDPI style. Names of the statistic tests should be included in the other abbreviations in order. P-values are unnecessary too long.
|
The table has been structured in accordance with MDPI requirements. The names of the statistical tests have been placed in the other abbreviations in order. P-values have been abbreviated. |
|
10 |
5. The following articles should be upgraded in the List of references and their content discussed: Fang, C. Y., Tsai, J. L., Li, G. S., Lien, A. S., & Chang, Y. J. (2020). Effects of Robot-Assisted Gait Training in Individuals with Spinal Cord Injury: A Meta-analysis. BioMed research international, 2020, 2102785. https://doi.org/10.1155/2020/2102785 Li R, Ding M, Wang J, et al. Effectiveness of robotic-assisted gait training on cardiopulmonary fitness and exercise capacity for incomplete spinal cord injury: A systematic review and meta-analysis of randomized controlled trials. Clinical Rehabilitation. 2023;37(3):312-329. doi:10.1177/02692155221133474 Anas R. Alashram, Giuseppe Annino, Elvira Padua, Robot-assisted gait training in individuals with spinal cord injury: A systematic review for the clinical effectiveness of Lokomat, Journal of Clinical Neuroscience, Volume 91, 2021, Pages 260-269, https://doi.org/10.1016/j.jocn.2021.07.019
|
The list of references has been updated to include the listed articles and their content discussed.
|
|
11 |
6. The Discussion should start from the short presentation of the most important findings.
|
As suggested, at the onset of the discussion we presented of the most important findings.
|
|
12 |
7. The editorial content of the manuscript leaves much to be desired, starting from the title page with affiliations and ending with references that are not ordered in accordance with the MDPI requirements.
|
The manuscript has been structured in accordance with MDPI requirements. |
|
13 |
8. Photographs presenting the methodological principles would increase the manuscript’s originality.
|
Photographs showing patients using the exoskeleton EXO-GT and Locomat Pro are included in the methods section of the manuscript. They are described as Figure 1 and Figure 2.
|
Round 2
Reviewer 1 Report
Most of my comments on the previous review have been address.
However, there is a clear typo in table 2, the final SCIM-III score for S0 can't be 8.83, based on the different provided and Figure 5.
Also I'd recommend providing a clear (non-tracked-changes) version in additional to the tracked changes version for final review.
Author Response
Dear Editor:
Thank you for your critical comments, which helped us greatly improve the manuscript.
The final SCIM-III score for S0 in table 2 has been corrected.
Additionally, minor linguistic corrections were made.
In the file below the attachment contains the manuscript in track changes mode - our latest corrections and manuscript with non-tracked-changes version.
Sincerely,
Justyna Frasuńska*(corresponding author)

Reviewer 2 Report
The Authors have improved the manuscript according to most of my comments.
Minor editing of English language required
Author Response
Dear Reviewer:
Thank you for your critical comments, which helped us greatly improve the manuscript.
The minor linguistic corrections were made.
Additionally, The final SCIM-III score for S0 in table 2 has been corrected.
In the file below the attachment contains the manuscript in track changes mode - our latest corrections.
Sincerely,
Justyna Frasuńska*(corresponding author)
